# AT101 [(-)-Gossypol] Selectively Inhibits MCL1 and Sensitizes Carcinoma to BH3 Mimetics by Inducing and Stabilizing NOXA

**DOI:** 10.3390/cancers12082298

**Published:** 2020-08-15

**Authors:** David J. Mallick, Alan Eastman

**Affiliations:** 1Department of Molecular and Systems Biology, Geisel School of Medicine at Dartmouth, Lebanon, NH 03756, USA; David.J.Mallick.GR@Dartmouth.edu; 2Department of Molecular and Systems Biology, and Norris Cotton Cancer Center, Geisel School of Medicine at Dartmouth, Lebanon, NH 03756, USA

**Keywords:** BCL2 family, apoptosis, BH3 mimetic, AT101, NOXA, MCL1

## Abstract

Anti-apoptotic BCL2 proteins are important targets for cancer therapy as cancers depend on their activity for survival. Direct inhibitors of MCL1 have entered clinical trials, although their efficacy may be limited by toxicity. An alternative approach may be to induce the pro-apoptotic protein NOXA which selectively inhibits MCL1 in cells. Many compounds originally proposed as inhibitors of the BCL2 family were subsequently found to induce the pro-apoptotic protein NOXA through the unfolded protein response. In the present study, we compared various putative BH3 mimetics across a panel of carcinoma cell lines and measured expression of NOXA protein and mRNA, as well as the kinetics of NOXA induction. We found that AT101 [(-)-gossypol] induces high levels of NOXA in carcinoma cell lines yet cells survive. When combined with an appropriate BCL2 or BCL-XL inhibitor, NOXA-dependent sensitization occurs. NOXA protein continues to accumulate for many hours after AT101 is removed, providing a window for administering these combinations. As MCL1 promotes drug resistance and overall survival, we propose that NOXA induction is an alternative therapeutic strategy to target MCL1 and either kill cancer cells that are dependent on MCL1 or sensitize cancer cells to other BCL2 inhibitors.

## 1. Introduction

Cancer cells often manipulate their internal apoptotic machinery such that survival is ensured [1]. The BCL2 family of proteins regulate apoptosis at the mitochondrial level through protein–protein interactions and are categorized into two groups: the anti-apoptotic multidomain proteins and the pro-apoptotic proteins that have either multiple domains or just the BH3 domain [2]. Many tumors become reliant on anti-apoptotic proteins for survival, hence these proteins have become attractive targets for therapy. Current small-molecule compounds called BH3 mimetics have been shown to be effective at targeting these proteins as they bind to the hydrophobic groove normally used to interact with the BH3 domain of the pro-apoptotic family members. Many BH3 mimetics have been developed. However, to date, only ABT-199 (venetoclax), which specifically targets BCL2, is FDA approved [3]. Further, while being effective in cancers that depend on BCL2 for survival, such as chronic lymphocytic leukemia (CLL), resistance to ABT-199, as well as other inhibitors of BCL2, can occur through upregulation of other anti-apoptotic proteins such as BCL-XL, BFL1 (BCL2A1), or MCL1 [4,5]. BH3 mimetics thought to antagonize these anti-apoptotic proteins have been reported. However, many of these compounds were discovered to act through alternative targets [6,7].

In tumors, MCL1 is frequently upregulated and plays a role in tumorigenesis, metastasis, and drug resistance, making it a highly valued therapeutic target [8,9,10]. Recently, several MCL1 inhibitors were developed that exhibit potent activity in directly antagonizing MCL1 in cells and in vivo: these include S63845, AZD5991 and AMG 176. All three compounds bind to MCL1 protein and dissociate pro-apoptotic BH3-only proteins; in addition, they inhibit MCL1 degradation which also depends on its BH3 pocket [11,12,13]. Phase 1 clinical development of these inhibitors is ongoing, although some trials have been suspended due to patient toxicities (ClinicalTrials.gov; NCT03465540, NCT03797261) [14]. MCL1 appears critical for proper development and maintenance of cardiac tissue which suggests direct inhibitors may have unexpected cardiac toxicities [15,16]. Therefore, there is still a critical need to find alternative mechanisms by which inhibition of MCL1 can be therapeutically achieved and safely combined with other BH3 mimetics to overcome drug resistance.

Many of the reported BH3 mimetics designed to antagonize MCL1 were found instead to induce NOXA through the unfolded-protein response (UPR) pathway [17,18]. In addition, apoptotic events caused by these compounds, whether through single-agent activity or through combination with another BH3 mimetic, were determined to be NOXA dependent. NOXA is a BH3-only pro-apoptotic protein that has natural selectivity for MCL1; thus, although not directly binding to MCL1, these compounds were still able to sensitize cancer cells to other BH3 mimetics [19,20]. This suggests that NOXA induction could be an effective means to antagonize MCL1 activity in cancer.

We compared various putative BH3 mimetics previously studied by our laboratory and found that AT101 induces NOXA across a wide variety of hematopoietic and epithelial cancer cell lines. AT101 [(-)-gossypol] is the active isomer of gossypol, a polyphenolic dialdehyde derived from natural cottonseed originally intended as an anti-fertility agent and administered to more than 10,000 male subjects [21]. AT101 was subsequently described as a BH3 mimetic having high affinity for multiple anti-apoptotic proteins and was used as such in phase I/II clinical trials [22,23,24]. However, evidence showing AT101 as a bona fide BH3 mimetic was based on in vitro experiments that did not fully explore its effects in cells; hence these clinical trials failed to show any efficacy with AT101. Our laboratory has shown that AT101, and gossypol, have apoptotic activity in leukemia and lymphoma cell lines that is NOXA dependent, whether as a single-agent or in combination with other BH3 mimetics, such as ABT-199 [19,25].

Here, we demonstrate that AT101 also induces NOXA in carcinoma lines, and that the mechanism by which NOXA is induced is similar to leukemia and lymphoma lines. Carcinoma cells tolerate high levels of NOXA but are sensitized to other BH3 mimetics in a NOXA-dependent manner. In addition, NOXA protein continued to accumulate for many hours after AT101 was removed, which kept cells primed for apoptosis upon addition of another BH3 mimetic. We propose that NOXA induction through AT101 is an alternative therapeutic strategy for targeting MCL1 and that NOXA-inducing compounds may have more utility in the clinic than realized.

## 2. Results

### 2.1. Comparison of NOXA Induction across Cell Lines

While many of the putative BH3 mimetics were shown to induce NOXA in leukemia and lymphoma cell lines, it was unknown whether these compounds would induce NOXA in carcinoma cell lines, or which compound is the most potent. We chose three mimetics whose mechanisms of NOXA induction were established by our laboratory: AT101, S1 and UMI-77 [18,20,25]. In addition, we have shown that microtubule-disrupting agents can also induce NOXA, and used BNC105 as a representative agent [26]. We selected a variety of carcinoma cell lines based on reports of their differential dependence on BCL2 family members [27,28,29]. Each compound was used at a concentration that was previously shown to induce NOXA in leukemia cells and all incubations were for 6 h (Figure 1A). We compared five carcinoma cell lines and one hematopoietic cell line and quantified the amount of NOXA protein (Figure 1B) and mRNA (Figure 1C) induced by each compound. None of the agents induced apoptosis during incubation as reflected in the lack of cleavage of PARP, a caspase 3 substrate. While the induction of NOXA protein was not uniform for each compound across cell lines, we found that AT101 consistently induced NOXA protein and mRNA and was generally the most potent of agents in this regard. We decided to focus on AT101 in subsequent studies as it has the most extensive clinical testing to date compared to the other compounds.

### 2.2. AT101 Induces NOXA in Carcinoma through Ca^2+^-Dependent ER Stress

In leukemia cells, AT101 was found to induce NOXA by the UPR through a Ca^2+^-dependent mechanism [25]. The UPR involves induction of ATF3 and ATF4 which heterodimerize and bind to the NOXA promoter, subsequently increasing NOXA mRNA and protein [30]. We determined whether this was the same mechanism by which AT101 induced NOXA in carcinoma lines. In HCT116 cells, AT101 induced ATF3, ATF4 and NOXA protein at higher concentrations after a 6 h incubation (Figure 2A), and these same concentrations also increased NOXA mRNA (Figure 2B).

NOXA induction by AT101 was previously reported to require activation of phospholipase A2 (PLA2) and Ca^2+^ influx to initiate the UPR in leukemia and lymphoma cells [25]. Incubation with aristolochic acid, an inhibitor of PLA2, decreased NOXA induction in HCT116 cells (Figure 2C). Incubation with the intracellular Ca^2+^ chelator BAPTA AM also prevented AT101-mediated induction of both NOXA protein (Figure 2D) and mRNA (Figure 2E). In addition, we observed that 24 h incubation with AT101 induced greater NOXA protein compared to 6 h (Figure 2F). As NOXA is a p53 target gene [31], we determined whether AT101-induced NOXA was p53 dependent by incubating p53-null HCT116 cells with AT101 (Figure 2G). We found that the level of NOXA induction in p53-null cells was comparable to p53-wt HCT116 cells, suggesting that the NOXA induction is p53-independent. This finding is further supported by the panel in Figure 1, in which NOXA is induced in all six cell lines despite only one cell line (HCT116) containing functional p53. Thus, the mechanism by which AT101 induces NOXA is the same for both leukemia and carcinoma.

Interestingly, we observed accumulation of MCL1 protein at 10 µM AT101 (Figure 2A), with no corresponding increase in MCL1 mRNA (Figure 2H), suggesting a post-translational modification. Further, chelation of Ca^2+^ prevented accumulation of MCL1 protein (Figure 2D) while silencing of NOXA through siRNA decreased MCL1 protein (Figure 2I). NOXA has been reported to bind and mark MCL1 for degradation by the proteasome using a degron sequence on its C-terminal tail [32]. Whether AT101-induced NOXA stabilizes MCL1 more than unoccupied MCL1 in the short term remains unknown.

### 2.3. AT101 Sensitizes Carcinoma Cells to Other BH3 Mimetics through NOXA

AT101 alone induced little apoptosis in any of the cell lines despite induction of NOXA (Figure 1A). We and others have shown that NOXA-inducing compounds can sensitize cancer cells to other compounds, such as inhibitors of BCL2 and BCL-XL [18,33,34]. We sought to determine whether AT101 could sensitize carcinoma cells to other BH3 mimetics. HCT116 cells incubated with AT101 were sensitized to ABT-263, an inhibitor of both BCL2 and BCL-XL (Figure 3A). Selective inhibition of BCL-XL (A1155463) induced slightly greater sensitization, whereas selective inhibition of BCL2 (ABT-199) did not sensitize these cells, suggesting that HCT116 cells rely on BCL-XL and MCL1 for survival. Similarly, SW620 cells, another colorectal carcinoma line, responded to the inhibitor of BCL-XL but not BCL2 when combined with AT101. In contrast, MiaPaca2 cells were sensitized to all three BH3 mimetics when combined with AT101, suggesting that inhibition of either BCL2 or BCL-XL with MCL1 in this cell line is sufficient to induce apoptosis (Figure 3B). Knockdown of NOXA with siRNA (Figure 3C) as well as caspase-inhibition (Figure 3D) protected HCT116 cells from sensitization to A1155463 confirming that AT101-induced sensitization is both NOXA and caspase dependent.

### 2.4. AT101 Is Comparable to Direct MCL1 Inhibitor AZD5991

Since we are interested in using AT101 as an alternative antagonist of MCL1 through NOXA, we compared AT101 with AZD5991, an established direct MCL1 inhibitor [12]. Incubation with AZD5991 resulted in the accumulation of MCL1 protein without induction of NOXA protein, consistent with a bona fide MCL1 inhibitor (Figure 4A). For comparisons, we used a concentration of AZD5991 which resulted in the largest accumulation of MCL1 protein without triggering apoptosis in order to fully saturate MCL1. We found that AZD5991 induced MCL1 to a higher level at 24 h than at 6 h and that AT101 induced NOXA to a higher level at 24 h than at 6 h (Figure 2F). Therefore, in these experiments we incubated cells concurrently for 24 h. HCT116 cells were sensitized to ABT-263 by both AT101 and AZD5991, with AT101 yielding slightly greater sensitization as indicated by cleaved PARP (Figure 4B). Using the specific BCL-XL inhibitor A1155463, both compounds sensitize HCT116 cells down to 0.6 µM A1155463. Similarly, PC3 cells responded to both AT101 and AZD5991 when combined with ABT-263, with AZD5991 causing full PARP cleavage (Figure 4C). In SW620 cells, 0.1 µM AZD5991 was sufficient to sensitize cells to A1155463, and this sensitization was slightly greater than that observed with AT101 (Figure 4D). These results suggest that NOXA induction is generally comparable to direct MCL1 inhibitors in antagonizing MCL1 protein and causing sensitization.

### 2.5. AT101 Promotes Long-Term Stabilization of NOXA Protein

To optimize cell killing, we investigated the kinetics of AT101-mediated induction of NOXA in four cell lines and found that NOXA protein began to accumulate around 4 h (Figure 5A). Interestingly, MCL1 protein also accumulated around the same time points as NOXA (as also observed in Figure 2A). Given that AT101 induces both NOXA and MCL1 protein, we wondered whether AT101 stabilized NOXA protein and whether this is through stabilization of the NOXA–MCL1 complex.

AT101 has been administered to patients orally with a plasma half-life of 2 h [22,24,35]. We therefore approximated this exposure by following the expression of NOXA after removal of AT101 at 6 h. Surprisingly, there was a dramatic increase in NOXA protein peaking at 12–24 h (Figure 5B). MCL1 protein also accumulated and correlated with the accumulation of NOXA protein, suggesting stabilization of the complex. NOXA mRNA decreased after AT101 removal, but with a secondary peak at 48 h. Interestingly, a small amount of PARP cleavage occurred 24 h after AT101 removal, but was not observed at 48 h, suggesting that cells undergoing apoptosis had disintegrated by 48 h, leaving only surviving cells. Measurement of NOXA half-life using cycloheximide indicated that the NOXA half-life was around 0.5 h consistent with prior reports (Figure 5C) [32]. However, following incubation with AT101, NOXA protein was much more stable with a half-life of around 2 h. Thus, AT101 not only induced NOXA protein, but also promoted its post-translational stabilization. We also observed stabilization of MCL1 protein following incubation with AT101 with a half-life of around 2 h. Additionally, we compared the expression of NOXA after the removal of each compound used in the panel (Figure 1) and found that AT101 had both the highest and most stable expression of NOXA (Figure 5D). While there was also an increase in NOXA following a 6 h incubation with S1 and UMI-77, there was no increase after removal of BNC105 suggesting the increase is related to the activation of the UPR. We also found that AT101 increased NOXA in NB4 cells far more than S1 24 h after removing drug (Figure 5E), despite S1 having a higher NOXA expression after 6 h (Figure 1B).

If NOXA protein is stabilized for long periods, then it could keep cells primed for apoptosis as long as it remains induced. We tested this hypothesis by incubating HCT116 cells with AT101 and A1155463 either concurrently or with a delayed schedule (Figure 5F). Based on cleaved PARP, incubation for 6 h with A1155463 starting 18 h after washout of AT101 resulted in greater sensitization than concurrent treatment, suggesting there is a therapeutic window where AT101 exhibits enhanced sensitization long after its removal.

### 2.6. AT101 Primes Cells for Sensitization through NOXA Expression

The efficacy of BH3 mimetics in killing cancer cells may be predicted by determining how primed cancer cells are for apoptosis. The priming concept suggests that cancer cells can more readily undergo apoptosis by BH3 mimetics if they have a large pool of pro-apoptotic proteins sequestered by anti-apoptotic proteins at the mitochondria [36]. BIM is a BH3-only protein that binds to all anti-apoptotic members of the BCL2 family, and NOXA is able to release BIM from MCL1 [17]. This released BIM would then bind BCL2 and BCL-XL and release BAD, a BH3-only protein that binds BCL2 and BCL-XL but not MCL1. The release of BAD can be observed by its translocation from the mitochondria to the cytosol using a digitonin assay. Given that AT101 induces NOXA that results in sensitization, we determined if AT101 primes cells through NOXA induction. When HCT116 cells were incubated with AT101, we observed no translocation of BAD from pellet fractions into the cytosol, suggesting that neither AT101 itself nor its NOXA induction was displacing BAD from BCL2 or BCL-XL (Figure 6A). As a positive control, we showed that ABT-263 (“A”) did displace BAD from the mitochondria fraction. Since NOXA was unable to release BAD, this suggests that NOXA is either fully sequestered by MCL1 or displaces few other BH3 proteins that are insufficient to disrupt the binding partners of BCL2 or BCL-XL; this explains why cells are tolerant of NOXA induction. Incubation of HCT116 cells with AZD5991 was also unable to release BAD (Figure 6B), again suggesting that there are insufficient pro-apoptotic BH3-only proteins displaced from MCL1 to disrupt the balance in favor of apoptosis. Consistent with these observations, immunoprecipitation of MCL1 showed that MCL1 binds most of NOXA both basal and after induction with AT101 but does not disrupt the small fraction of MCL1-bound BIM (Figure 6C). It is likely that MCL1 is saturated by the induced NOXA, leaving no MCL1 free to bind BIM displaced from BCL2 or BCL-XL upon incubation with BCL2 or BCL-XL inhibitors, thereby resulting in apoptosis when AT101 is combined with these other BH3 mimetics.

## 3. Discussion

Development of small-molecule inhibitors of the anti-apoptotic BCL2 subfamily has become an important strategy to push cancer cells towards apoptosis. So far, only ABT-199 (venetoclax) has achieved FDA approval for certain indications where BCL2 inhibition is effective. While a successful inhibitor of MCL1 is close at hand with compounds like S63845, AMG 176 and AZD5991 entering clinical trials, alternative therapies must continually be developed as resistance is inevitable. NOXA is an effective, natural antagonist of MCL1 that can both kill MCL1-dependent cells and synergize with other compounds. Many putative BH3 mimetics were misclassified as direct MCL1 inhibitors, yet they were still able to sensitize cancer cells to other compounds because of their ability to induce NOXA, revealing their therapeutic potential [17,18]. Here, we show that AT101, a previously misclassified BH3 mimetic enantiomer of gossypol, induces NOXA in multiple carcinoma cell lines and also sensitizes in a NOXA-dependent manner. Given the recent cardiac toxicities seen with MCL1 inhibitors, NOXA induction might be a suitable alternative to target MCL1 in cancer cells [14].

Our laboratory previously demonstrated that AT101 induced NOXA in leukemia and lymphoma cells through a mechanism involving PLA2 activation, Ca^2+^ influx and the unfolded protein response [25]. This mechanism is the same for carcinoma cells treated with AT101. Despite inducing NOXA, AT101 induced little apoptosis in any of the carcinoma cell lines tested, indicating a tolerance for high levels of NOXA. Induction of NOXA has been suggested to prime cells for apoptosis, although these experiments were conducted mainly in leukemia and lymphoma cells [37,38]. Considering that cells are sensitized by AT101 to other compounds, it is most likely that the induction of NOXA alone is not sufficient to kill cells, but instead primes cells to undergo apoptosis more easily by another compound.

The level of sensitization seen with AT101-induced NOXA is comparable to, if not better than, that seen with AZD5991, a direct MCL1 inhibitor. Interestingly, we observed that all carcinoma lines appeared equally sensitive to AT101-mediated sensitization to BCL-XLi, yet there was variable sensitivity to AZD5991-mediated sensitization, with some cells requiring a higher concentration of AZD5991 to achieve sensitivity comparable to AT101. Perhaps in cases where some cell lines require more AZD5991 to become sensitized to BCL-XLi or other compounds, AT101-mediated NOXA induction would be better. There are multiple agents that have been reported to induce NOXA although most are through the UPR or p53. Pharmacological inhibition of the proteasome upregulates NOXA expression through both stability of its transcription factors, such as p53, and activation of the UPR [39,40]. Oxidative stress can lead to NOXA induction through generation of ROS, as reported for S1, although the mechanisms by which ROS generates NOXA is probably also mediated through the UPR [20]. DNA-damaging agents, such as UV radiation or topoisomerase inhibitors, are also known to induce NOXA but require functional p53 to do so [41].

AT101 is able to not only induce NOXA mRNA and protein in various carcinoma cell lines but also promote post-translational stabilization of NOXA protein up to 24 h after its removal. How AT101 stabilizes NOXA remains to be determined, but considering the concurrent increase in MCL1, and co-immunoprecipitation of MCL1 and NOXA, it is plausible that this results from increased stability of the NOXA–MCL1 complex. NOXA is reported to contain a degron sequence on its C-terminal tail that is exposed when bound to MCL1, marking the complex for degradation by the proteasome [32]. The NOXA–MCL1 complex has been shown to be degraded through proteasomal turnover regulated by MARCH5 and MTCH2 and that stabilization of this complex hinders the pro-survival role of MCL1 despite its accumulation [42]. Alternatively, gossypol, a precursor of AT101, has been shown to inhibit neddylation of both SAG-CUL5 and RBX1-CUL1, resulting in accumulation of both NOXA and MCL1 in multiple cell lines [43]. Whether these pathways are dysfunctional upon incubation with AT101, or whether the proteasome is overtaxed and unable to degrade the marked complexes remains unknown.

The fact that AT101 is not a pan-BCL2 inhibitor as originally anticipated, and that cells tolerate high levels of NOXA may explain why AT101 failed in previous clinical trials. AT101 was originally designated a BH3 mimetic targeting multiple anti-apoptotic proteins, including BCL2, BCL-XL and MCL1 [44]. These studies used in silico and in vitro models to determine binding affinity, which has been extensively shown to be an inaccurate predictor for BH3 mimetic efficacy in cells [6,7]. AT101 might have induced NOXA in patient tumors, but if it was assumed to inhibit all anti-apoptotic proteins directly, the lack of apoptosis would lead to the conclusion that it had no single-agent efficacy. We have shown that AT101 is most effective as a sensitizer of cancer cells and that NOXA is a critical component of this sensitization. Further, AT101 is able to stabilize NOXA protein such that high levels remain in cells long after AT101 is removed. We believe that AT101 should be reconsidered as a viable clinical tool, with the understanding that (1) AT101 must be combined with another compound that synergizes with NOXA induction; (2) pharmacokinetic (PK) measurements will not predict AT101 efficacy in tumors as the protein continues to increase long after AT101 is removed; and (3) the induction of NOXA in the tumor should be assessed as a clinical biomarker of AT101.

This work complements our earlier suggestion that induction of NOXA could be valuable in the clinic. We previously noted that microtubule inhibitors also induce NOXA and are currently evaluating one in a clinical trial (ClinicalTrials.gov; NCT03454165) [26,45]. We now realize that, not only is AT101 more potent, but also that it has great efficacy in carcinoma cell lines. The idea that induction of NOXA is a valid strategy has more recently been suggested by Jin et al. showing 5-azacitidine can also induce NOXA and sensitize acute lymphocytic leukemia cells to venetoclax [38]. Two clinical trials of this combination are ongoing, although the possible induction of NOXA was apparently not initially realized (ClinicalTrials.gov; NCT03466294, NCT03573024). Additionally, pharmacokinetic analysis of AT101 from previous clinical trials suggest that the concentrations of AT101 used throughout this paper (up to 10 µM) are close to plasma concentrations achieved in patients. Patients receiving 40 mg AT101 orally achieved a peak plasma concentration of about 4 µM with a half-life of 3.44 h [46]. In addition, AT101 was previously administered on a daily schedule, and toxicities occurred after multiple doses, whereas our results suggest less frequent administration may be effective, such that higher concentrations may be tolerated. Our data with AT101 suggest that the next step should be a proof-of-concept clinical trial to determine whether AT101 can induce NOXA protein in patients at doses that are tolerated.

## 4. Materials and Methods

### 4.1. Cell Culture and Reagents

Human acute promyelocytic leukemia NB4 cells were a gift from Dr. Ethan Dmitrovsky (Frederick National Laboratory, Frederick, MD, USA). HCT116 p53-null cells were obtained from Dr. Burt Vogelstein (Johns Hopkins University, Baltimore, MD, USA). Other cell lines were obtained from the American Type Culture Collection (Manassas, VA, USA). All cell lines were maintained in RPMI 1640 containing 10% (*v/v*) fetal bovine serum and 1% (*v/v*) antibiotic-antimycotic solution. Cell lines were tested for mycoplasma using the MycoAlert^TM^ Mycoplasma Detection Kit (Thermo Fisher, Waltham, MA, USA). Cell lines positive for mycoplasma contamination were treated through use of Plasmocure^TM^ (Invivogen, San Diego, CA, USA).

ABT-199 and A1155463 were provided by AbbVie (North Chicago, IL, USA). ABT-263, AT101 and UMI-77 were purchased from Selleckchem (Houston, TX, USA). S1 was synthesized according to a previously published method [47]. BNC105 was obtained from Bionomics (Thebarton, South Australia). AZD5991 was purchased from Chemgood (Kowloon, Hong Kong). The intracellular Ca^2+^ chelator BAPTA AM and pan-caspase inhibitor Q-VD-OPH were purchased from Cayman Chemicals (Ann Arbor, MI, USA). Cycloheximide was obtained from Sigma-Aldrich (St. Louis, MO, USA). Hoechst 33,258 was purchased from Molecular Probes (Thermo Fisher). Where indicated, Q-VD-OPH or BAPTA AM were added to cells 1 h prior to treatment at a concentration of 10 µM.

### 4.2. Immunoblot Analysis

Cells were lysed in 10 mM Hepes (pH 7.5), 300 mM KCl, 1% NP-40 and protease/phosphatase inhibitor mixture. Protein concentration of lysates was measured using a Bradford protein assay from Bio-Rad (Hercules, CA, USA) before an equal volume of 2× urea buffer (8 M urea, 100 mM Tris (pH 6.8), 10% β-mercaptoethanol, 4% SDS, 0.01% bromophenol blue, and protease/phosphatase inhibitor cocktail) was added to each lysate. Lysates were then heated on a 95 °C heat block for 5 min. Proteins were separated by SDS-PAGE (6%, 8% 10%, 15% or 18%) and transferred to a polyvinylidene difluoride (PVDF) membrane (Millipore, Burlington, MA, USA). All membranes were blocked in 5% non-fat milk in Tris-buffered saline then incubated in either Odyssey^®^ Blocking Buffer (TBS) solution with 0.2% Tween20, or 5% non-fat milk in Tris-buffered saline, 0.05% Tween20 with the appropriate primary antibody overnight. Subsequently, membranes were washed in Tris-buffered saline, 0.05% Tween20 and incubated with secondary antibody conjugated to DyLight^TM^ 800 4× PEG fluorescent dye (Cell Signaling Technology, Danvers, MA, USA). Proteins were visualized by the Odyssey CLx Imager (LI-COR, Lincoln, NE, USA). Vinculin was used as loading controls in Western blots. When comparing cell lines, the same protein concentration was loaded on each gel.

Antibodies were obtained from the following sources: Cell Signaling: ATF4 (D4B8), BIM (2933), MEK1/2 (9122), PARP (9532), PUMA (12450), and Tom20 (42406); BD Biosceinces, San Jose, CA, USA: MCL1 (559027); Santa Cruz Biotchnology, Dallas, TX, USA: ATF3 (sc-188), BAD (sc-8044), and vinculin (sc-73614); Sigma-Aldrich: Actin-HRP (A3854), and NOXA (OP180). Secondary antibodies were purchased from Cell Signaling (Danvers, MA, USA). For immunoprecipitations: Santa Cruz Biotechnology: MCL1 (sc-819) and IgG (sc-2025).

### 4.3. Immunoprecipitation Assay

Cells were lysed in F-buffer (10 mM Tris pH 7.05, 50 mM NaCl, 30 mM Na pyrophosphate, 5 mM ZnCl, 10% glycerol, 0.5% Triton X-100 and protease/phosphatase inhibitors) for 10 min on ice, then centrifuged at 13,000 rpm for 15 min. A portion of the supernatant was collected and resuspended in 2× Laemmli sample buffer containing 100 mM DTT and boiled for 10 min. The supernatant was collected and antibody [2.0 µg; MCL1 or IgG] was added and mixed for 1 h at 4 °C. Pre-washed magnetic beads (Classic Magnetic IP/Co-IP Kit; Pierce, 8804) were added and incubated on a rotator at 4 °C overnight. The supernatant (flow-through, FT) was recovered, and the immunoprecipitate (IP) was washed and resuspended in 2× Laemmli sample buffer containing 100 mM DTT. Lysates were then boiled for 10 min before loading onto SDS-PAGE. Equivalent aliquots of supernatant and immunoprecipitate were subjected to Western blotting performed for the proteins indicated.

### 4.4. Chromatin Condensation Assay

Chromatin condensation is an early and quantifiable hallmark of apoptosis that occurs at about the same time as the cleavage of PARP. Cells were harvested and incubated with 2 µg/mL Hoechst 33,258 for 20 min at 37 °C and visualized with a fluorescent inverted microscope. At least 200 cells were scored from each sample and data were expressed as the percentage of cells with condensed chromatin.

### 4.5. Digitonin Permeabilization (BAD Assay)

Separation of membrane/organelle fraction from cytosol was performed based on previously published methods [48]. Cells were incubated with digitonin buffer (8.75 µg digitonin/10^6^ cells, 75 mM NaCl, 1 mM NaH_2_PO_4_, 8 mM Na_2_HPO_4_, 250 mM sucrose and protease/phosphatase inhibitor mixture) for 2 min on ice. Cells were then centrifuged at 12,500 rpm for 1 min at 4 °C. The supernatant was transferred and supplemented with an equal volume of 2× urea buffer. The pellet was resuspended in an equal volume of digitonin buffer and 2× urea buffer. The samples were then boiled for 5 min prior to Western blotting.

### 4.6. RT-qPCR Analysis

Total mRNA was extracted using TRIzol reagent (Sigma-Aldrich). Complimentary DNA (cDNA) was synthesized using iScript^TM^ cDNA synthesis kit (Bio-Rad, Hercules, CA, USA). Final RNA concentrations were measured by absorbance at 260 nM and quality was assessed using an A260/280 ratio. cDNA was prepared using 400 ng of total RNA in 20 µL reverse transcription reaction with the iScript^TM^ cDNA Synthesis Kit (Bio-Rad) according to the manufacturer’s protocol. Further details are described in Appendix A.

RT-qPCR reactions were performed in a 25 µL reaction containing 0.5 µL of diluted cDNA, 12.5 µL 2× SYBR Green MasterMix (Bio-Rad) or 2× EvaGreen MasterMix (Biotium, Fremont, CA, USA) and 0.5 µL of each forward and reverse primer at final concentrations of 200 nM. All qPCR reactions were run in triplicate using TempAssure^TM^ 0.2 mL PCR 8-tube strips and optical caps (USA Scientific, Ocala, FL, USA) on a CFX96 Real-Time System (Bio-Rad). Amplification conditions were adapted to each primer based on their primer melting temperature (T_m_) but mainly consisted of the initial denaturation step at 95.0 °C for 10 min, followed by 40 cycles of 30 s at 94.0 °C, primer melting temperature for 30 s and 72.0 °C for 1 min. Afterwards, a melting curve was generated for each primer set to confirm the presence of a single uniform peak, which indicates on-target primer efficacy. Results were analyzed in Microsoft Excel by comparative C(t) (ΔΔC(t)) using GAPDH as a normalization control and equations previously published [49]. Results were then analyzed and compiled for presentation in GraphPad Prism (GraphPad Software, San Diego, CA).

The following primers (5′-3′) were provided by Integrated DNA Technologies (Coralville, IA, USA): GAPDH: (F) CTCAGACACCATGGGGAAGGTGA and (R) ATGATCTTGAGGCTGTTGTCATA; MCL1: (F) TCAAAAACGAAGACGATGTGA and (R) CAAAGGCACCAAAAGAAATGA; NOXA: (F) AAGAAGGCGCGCAAGAAC and (R) TCCTGAGCAGAAGAGTTTGG.

### 4.7. Cell Transfection and siRNA Knockdown

Knockdown of NOXA was performed using Lipofectamine^®^ RNAiMAX reagent according to the manufacturer’s protocol. The following siRNAs were obtained from Ambion (Austin, TX, USA): NOXA (PMAIP1) [s10709], (AGUCGAGUGUGCUACUCAAtt) and the non-targeting siRNAs (Silencer^®^ Select Negative Control No. 1 and No. 2).

### 4.8. Statistical Analysis

For qPCR analysis, all data are presented as a fold change relative to the untreated condition ± range. The unpaired, two-tailed Student’s *t*-test was applied to the dC(t) values to determine the statistical significance between experiment groups. All other data are presented as the mean ± stdev and the Student’s *t*-test was used to determine the statistical significance between experimental groups.

## 5. Conclusions

AT101 induces a high level of NOXA protein in carcinoma cell lines and sensitizes them to BCL-XL and/or BCL2 inhibitors in a NOXA-dependent manner. The sensitization seen with AT101 is comparable to direct MCL1 inhibition, suggesting that indirect inhibition of MCL1 through NOXA is a viable alternative to MCL1-targeted therapy. In addition, AT101 stabilizes NOXA protein such that cells remain sensitized up to 24 h after removal of AT101. This suggests that the oral administration of AT101 would be suitable to induce NOXA and achieve a window of time where the tumor becomes sensitized to another compound administered at a later time. We urge others to reexamine AT101 as well as related NOXA inducing compounds as an alternative treatment strategy as the potential efficacy of NOXA induction has yet to be realized.

## Figures and Tables

**Figure 1 cancers-12-02298-f001:**
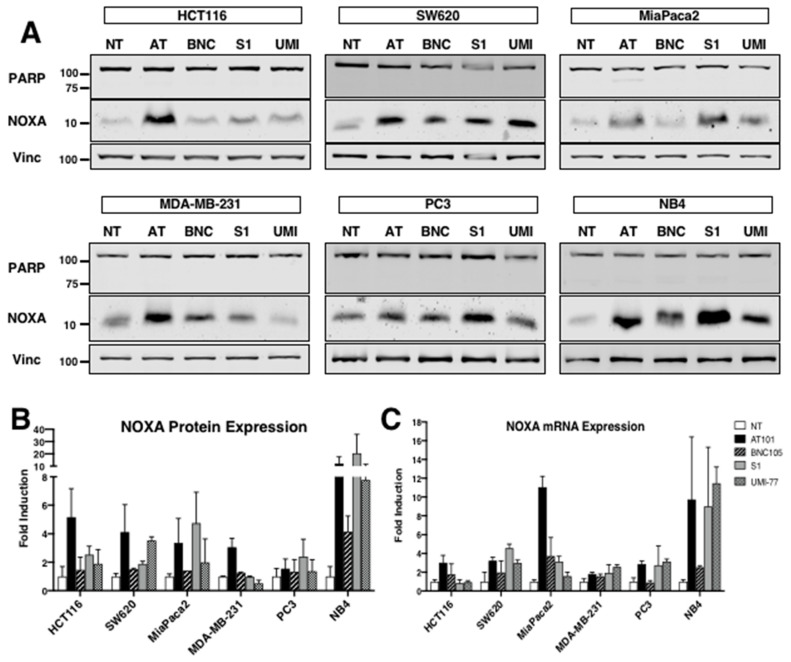
Induction of NOXA by BH3 mimetics. (**A**) Six cell lines were incubated with either 10 μM AT101 (AT), 1 μM BNC105 (BNC), 20 µM S1 (S1) or 30 µM UMI-77 (UMI) for 6 h and assessed for PARP cleavage as a marker of apoptosis, NOXA and vinculin expression by Western blotting. NB4 is a hematopoietic cancer cell line while the others are carcinoma cell lines. (**B**) NOXA signal intensity was measured with the Odyssey CLx Imager (LI-COR) for each cell line and expressed relative to the untreated cell lysates. Error bars represent range (*n* = 2) except for MiaPaca2 BNC105 (*n* = 1). (**C**) NOXA mRNA from similarly incubated cells was quantified using qPCR. Error bars represent the range of fold-change values (*n* = 3).

**Figure 2 cancers-12-02298-f002:**
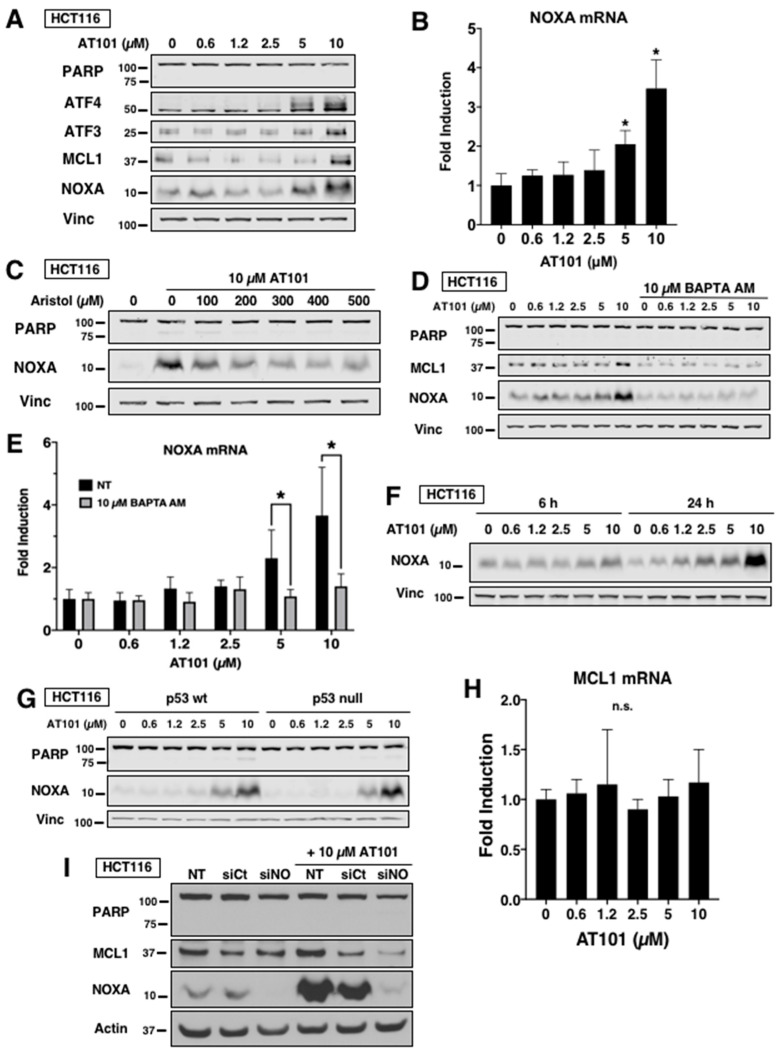
Mechanism of NOXA induction by AT101. (**A** and **B**) HCT116 cells were incubated with 0–10 µM AT101 for 6 h and analyzed for markers of UPR by Western blotting (**A**) and NOXA mRNA (**B**). Fold change was expressed relative to untreated cells. Error bars represent the range of fold-change values. Statistical analysis was applied to the dC(t) values (*n* = 3, * *p* < 0.05). (**C**) HCT116 cells were incubated with 0–500 µM of the PLA2 inhibitor aristolochic acid for 1 h and then 10 µM AT101 was added for 6 h. (**D** and **E**) HCT116 cells were incubated with the intracellular Ca^2+^ chelator BAPTA AM for 1 h and then AT101 was added for 6 h; cells were then analyzed for NOXA protein (**D**) or NOXA mRNA (**E**). Error bars represent the range of fold-change values and statistical analysis was applied to the dC(t) values (*n* = 3, * *p* < 0.05). (**F**) HCT116 cells were incubated with AT101 for 6 h or 24 h and then analyzed for NOXA protein. (**G**) p53 wt- or p53-null HCT116 cells were incubated with AT101 for 6 h and analyzed for NOXA protein. (**H**) HCT116 cells were incubated with AT101 for 6 h and MCL1 mRNA was quantified using qPCR. Error bars represent the range of fold-change values (*n* = 3, n.s. = no significance). (**I**) HCT116 cells were transfected with non-targeting siRNA (siCt) or siRNA against NOXA (siNO) then incubated with AT101 for 6 h and analyzed for PARP, NOXA and MCL1 protein.

**Figure 3 cancers-12-02298-f003:**
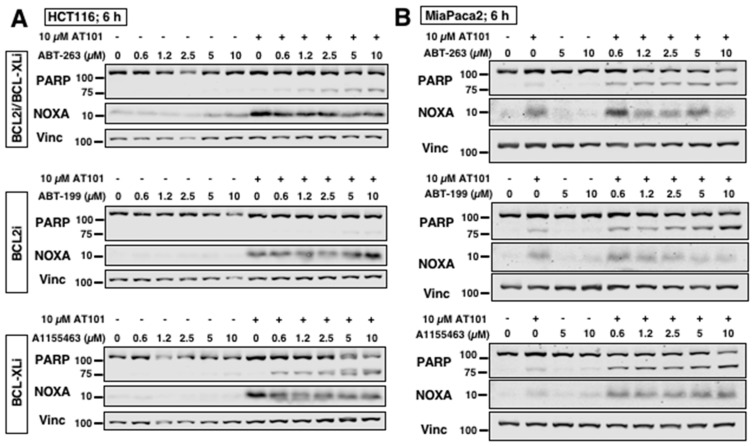
AT101 sensitizes carcinoma cells to other BH3 mimetics. (**A**) HCT116 cells and (**B**) MiaPaca2 cells were incubated with either ABT-263, ABT-199 or A1155463 alone or in combination with 10 µM AT101 for 6 h. (**C**) HCT116 cells were transfected with non-targeting siRNA (siCtrl) or siRNA against NOXA (siNOXA) then incubated with either 5 µM A1155463, 10 µM AT101 or both for 6 h. Apoptosis was assessed by PARP cleavage and chromatin condensation, expressed as the percentage of surviving cells (*n* = 3, error bars represent standard deviation, ** *p* < 0.01). (**D**) HCT116 cells were incubated with 10 µM QVD for 1 h and then with A1155463 alone or in combination with 10 µM AT101 for 6 h.

**Figure 4 cancers-12-02298-f004:**
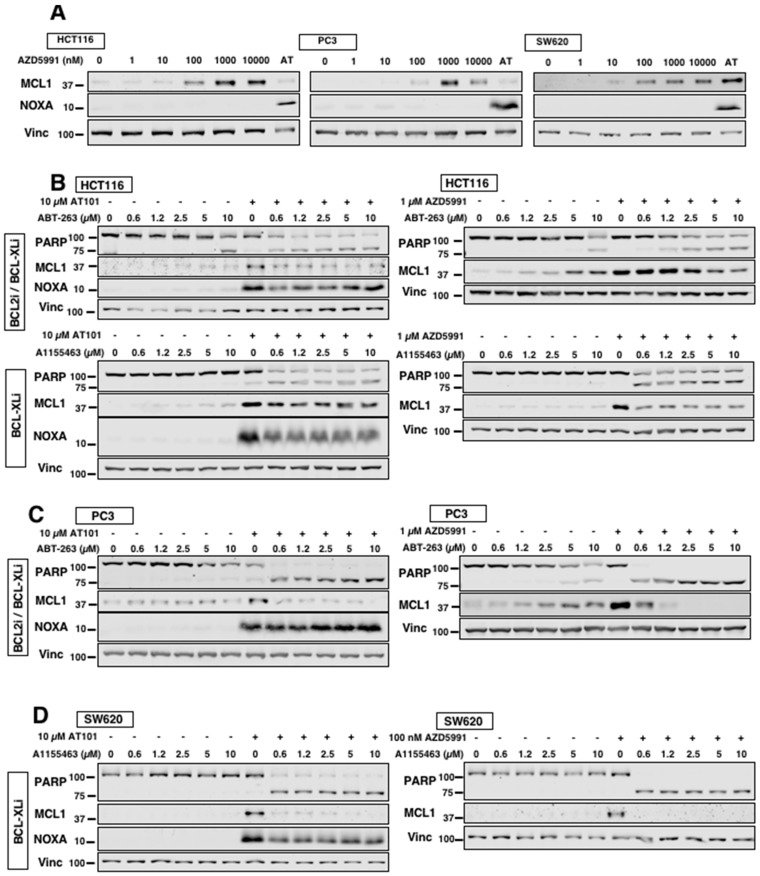
Comparison of AT101 to MCL1 inhibitor AZD5991. (**A**) The indicated cell lines were incubated with AZD5991 for 24 h and analyzed for NOXA and MCL1 protein expression. A 6 h incubation of AT101 (AT) was used as a positive control for NOXA induction. (**B–D**) The indicated cell lines were incubated with either 10 µM AT101 or 0.1–1 µM AZD5991 in combination with either ABT-263 or A1155463 for 24 h and analyzed for PARP cleavage, NOXA and MCL1 protein expression.

**Figure 5 cancers-12-02298-f005:**
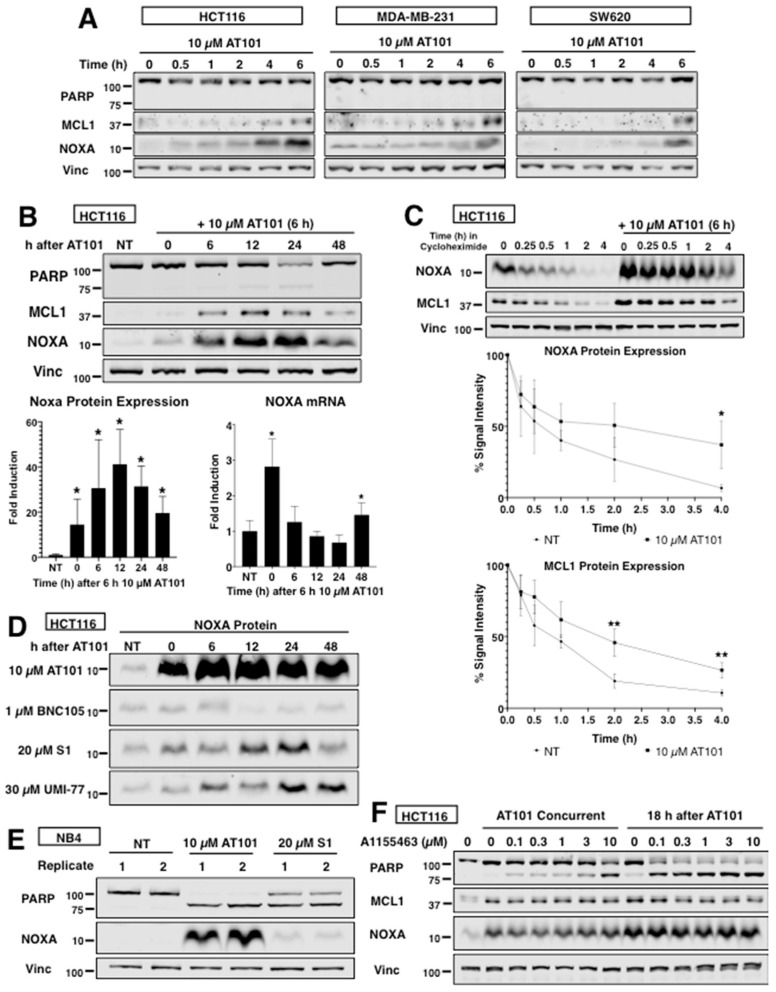
Kinetics of NOXA induction by AT101. (**A**) The indicated cell lines were incubated with 10 µM AT101 for 0–6 h and analyzed for NOXA and MCL1 protein expression. (**B**) HCT116 cells were incubated with 10 µM AT101 for 6 h, then washed and incubated for a further 6–48 h. Protein expression was calculated based on signal intensity normalized to vinculin and error bars represent the range of fold-change values (*n* = 3, * *p* < 0.05). NOXA mRNA was quantified by qPCR and error bars represent the range of fold-change values and statistical analysis was applied to the dC(t) values (*n* = 3, * *p* < 0.05). (**C**) HCT116 cells were incubated with 50 µg/mL cycloheximide for 0–4 h. Alternatively, cells were incubated with 10 µM AT101 for 6 h, then released into media for 18 h, then 50 µg/mL cycloheximide was added for an additional 0–4 h. NOXA and MCL1 protein expressions were calculated using signal intensities normalized to the 0 h lysate. Error bars represent standard deviation (*n* = 3, * *p* < 0.05, ** *p* < 0.01). (**D**) HCT116 cells were incubated with either 10 µM AT101, 1 µM BNC105, 20 µM S1 or 30 µM UMI-77 for 6 h, then washed and incubated for a further 6–48 h. (**E**) NB4 cells were incubated with either 10 µM AT101 or 20 µM S1 for 6 h, then washed and incubated for a further 24 h. (**F**) HCT116 cells were incubated with 10 µM AT101 for 6 h concurrently with A1155463, or with AT101 for 6 h followed at 18 h by a 6 h incubation with A1155463. Cell lysates were analyzed for PARP cleavage, NOXA and MCL1 protein expression.

**Figure 6 cancers-12-02298-f006:**
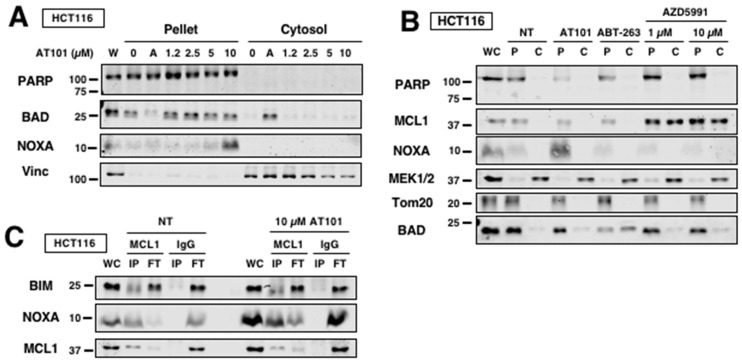
AT101 primes cells for sensitization through NOXA induction. (**A**) HCT116 cells were incubated with either 10 µM ABT-263 (“A”) or AT101 for 6 h and then permeabilized with digitonin and separated into pellet and cytosolic fractions. Translocation of BAD was measured by Western blotting. PARP and vinculin were used as controls for pellet and cytosol, respectively. (**B**) HCT116 cells were incubated with either ABT-263 (10 µM), AT101 (10 µM) or AZD5991 (1 or 10 µM) for 6 h and then permeabilized with digitonin and separated into pellet and cytosolic fractions. PARP and MEK1/2 were used as controls for pellet and cytosol, respectively. Tom20 was used as a mitochondrial marker. (**C**) HCT116 cells were incubated with 10 µM AT101 for 6 h and then subjected to MCL1 immunoprecipitation. Lysates were separated into immunoprecipitate (IP) and flow-through (FT) with IgG used as a negative control. BIM, NOXA and MCL1 expression were measured by Western blotting.

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
