# Peer review of "AT101 [(-)-Gossypol] Selectively Inhibits MCL1 and Sensitizes Carcinoma to BH3 Mimetics by Inducing and Stabilizing NOXA"

_cancers, 2020, doi:10.3390/cancers12082298_

Round 1
Reviewer 1 Report
The authors described the mechanistic background drug sensitization by AT101 in carcinoma cells. The overall data is good enough and worth publishing in Cancers.
Minor points
1. It will be good to add the MIQE checklist for qPCR experiment in supplementary info.
2. In p.14, 4.8 Statistical Analysis, lines between 442 and 453, I guess authors did not delete the template for the manuscript.
Author Response
- It will be good to add the MIQE checklist for qPCR experiment in supplementary info.
Answer: The MIQE checklist for all qPCR experiments has been filled out and included with this resubmission.
- In p.14, 4.8 Statistical Analysis, lines between 442 and 453, I guess authors did not delete the template for the manuscript.
Answer: The Statistical Analysis section has been edited such that the aforementioned template has been removed. (Line 496)
Reviewer 2 Report
This manuscript reports that AT101, a previously discredited so-called BH3-mimetic agent, induces the pro-apoptotic molecule NOXA in carcinoma cell lines, extending previous work that showed similar data in hematological malignancies (References 18, 20, and 25). They confirm that PLA2 and calcium influx similarly promote NOXA induction in carcinomas, echoing prior publications by this group in hematological malignancies. More interesting is that they demonstrate that AT101 acts very similarly to a bonafide BH3-mimetic AZD5991 in potentiating the activity of BCLxL inhibiting BH3-mimetics and that AT101 has long lasting effects in cells (even after washout) to promote NOXA induction.
The basic premise of the manuscript is solid as are most of the experiments. I have some concerns about the novelty of the core finding (which echoes Dr. Eastman’s previous work in hematological malignancies) and the lack of evidence that primary patient carcinomas will respond similarly, but the discovery does represent an advance.
Other concerns:
- NOXA and PUMA are in part p53 targets. While the authors conclude from Figure 2F that PUMA is only slightly induced, I would argue that it is induced, thus calling into question whether p53 plays a role in the AT101 induction of NOXA. This could be easily answered by testing the AT101 in p53-deficient HCT116 cells, which are widely available and well characterized.
- The induction of MCL1 by AT101 is suggested to be due to NOXA’s ability to stabilize MCL1 and prevent its proteasomal degradation. While the authors conclude that the mechanism remains unknown (line 140), it should be easy to test this concept. The authors can simply genetically ablate or silence NOXA (as they show in Figure 3C) and test whether MCL1 levels are stabilized. This would address this mechanism and at least provide an answer as to whether NOXA is required for the MCL1 stabilization.
- I am a little confused by the immunoprecipitation experiment shown in Figure 6C. The data indicate that most BIM is not bound to MCL1 (presumably other anti-apoptotics are binding the BIM), but that all NOXA is bound to MCL1 in normal cells. Then, when AT101 is added, NOXA is induced and while more NOXA is bound to MCL1, there is some free. This makes sense, but the BIM levels that are interacting with MCL1 do not seem to change. There is just as much BIM binding MCL1 and just as much unbound to MCL1. Then, how is cell death potentiated? I would have predicted from the authors’ conclusion that the induction of NOXA would outcompete BIM binding for MCL1 and lead to more BIM in the flow through. However, this is not what is observed. What is going on here? Is this true to for all carcinomas or only for HCT116?
Author Response
Reviewer #2
- NOXA and PUMA are in part p53 targets. While the authors conclude from Figure 2F that PUMA is only slightly induced, I would argue that it is induced, thus calling into question whether p53 plays a role in the AT101 induction of NOXA. This could be easily answered by testing the AT101 in p53-deficient HCT116 cells, which are widely available and well characterized.
Answer: In Figure 1A, the cell lines used in the panel all have a mutation that makes p53 defective except for HCT116 cells, which have functional p53. All cells lines in this panel induce NOXA when incubated with AT101, suggesting that p53 does not play a role in the AT101 induction of NOXA. This is now explained better in the text. Additionally, We have included data from an experiment where p53-null HCT116 cells were incubated with AT101 for 6 h and compared to p53-wt HCT116 cells (Figure 2G). The induction of NOXA in p53-null cells is comparable to p53-wt cells, further suggesting that AT101-induced NOXA is p53-independent. As the PUMA data was only obtained in the HCT116, and does not relate to induction of NOXA, we have now deleted that panel from Figure 2F. (Lines 129-130 and 149-153)
- The induction of MCL1 by AT101 is suggested to be due to NOXA’s ability to stabilize MCL1 and prevent its proteasomal degradation. While the authors conclude that the mechanism remains unknown (line 140), it should be easy to test this concept. The authors can simply genetically ablate or silence NOXA (as they show in Figure 3C) and test whether MCL1 levels are stabilized. This would address this mechanism and at least provide an answer as to whether NOXA is required for the MCL1 stabilization.
Answer: We have included data from an experiment where HCT116 cells were transfected with siNOXA and probed for NOXA and MCL1 after incubation with AT101 (Figure 2I). This data shows that when HCT116 cells are transfected with siNOXA and incubated with AT101, MCL1 protein is decreased, suggesting that NOXA stabilizes MCL1 protein. (Lines 132-134 and 157-158)
- I am a little confused by the immunoprecipitation experiment shown in Figure 6C. The data indicate that most BIM is not bound to MCL1 (presumably other anti-apoptotics are binding the BIM), but that all NOXA is bound to MCL1 in normal cells. Then, when AT101 is added, NOXA is induced and while more NOXA is bound to MCL1, there is some free. This makes sense, but the BIM levels that are interacting with MCL1 do not seem to change. There is just as much BIM binding MCL1 and just as much unbound to MCL1. Then, how is cell death potentiated? I would have predicted from the authors’ conclusion that the induction of NOXA would outcompete BIM binding for MCL1 and lead to more BIM in the flow through. However, this is not what is observed. What is going on here? Is this true to for all carcinomas or only for HCT116?
Answer: We have attempted to improve our explanation for this result shown in Figure 6C. The reviewer is correct that it appears when NOXA is induced by AT101 and binds MCL1, no BIM is displaced. However, it is now likely that there is no free MCL1 to bind any surplus BIM that would be released from BCL2/XL upon addition of a BCL2 or BCLXL inhibitor. Hence apoptosis ensues when this combination is used. This should be true for all carcinomas that express these BCL2 family proteins as the interactions among the family members is consistent throughout all cell lines. (Lines 294-297)
Reviewer 3 Report
This paper investeigated the role of AT101 in apoptosis induction in different cancer cell lines upon treatment with BH3 mimetics.
It was found that AT101 [(-)-gossypol] induces high levels of NOXA in carcinoma cell lines yet cells survive. Further studies showed that cancer cells treated by AT101become more sensitive to BH3 mimetics. More importantly, the paper demonstrated that the level of NOXA continues to accumulate for many hours after removing AT101, and this provided a window of time for administering other drugs.
The data are potentially interesting, however, the paper still has some flaws needs to be corrected.
- The author thought the AT101 was the most potent agent to induce the expression of NOXA, but from the results of Fig.1, S1 seems more efficient, especially in Miapaca2, PC3 and NB4. How did S1 induce the level of NOXA?
- In Fig.2D, BAPTA treatment increased the level of NOXA and MCL1 in HCT116. Did Aristol treatment also increase the level of MCL1?
- Both PUMA and NOXA are target genes of p53, however, Bort treatment only increased the level of PUMA but not NOXA. What’s the reason, and whether AT101 treatment could increase the level of p53?
- MCL1 detection seems required in Fig.3B-D.
- In Fig.1, AT101 could obviously increase the level of NOXA, but this increase seems not obvious in Fig.4A. Why?
- Fig.5B showed that the level of NOXA mRNA reached a secondary peak at 48h, but not the protein level. NOXA protein level needs to be detected for a longer period of time. And Fig.5A showed the result of AsPC1 which not mentioned in the early part of the paper, did AsPC1 mean PC3?
- Minor points.In Fig.4, (B through D) should be corrected to B-D (line 184).Line 289, “instead primes” should be corrected to “ instead of priming”.
- In Fig.5C, cycloheximide should be indicated.
- In first half part and latter part of the paper, the spelling of vinculin is inconformity.
Author Response
Reviewer #3
- The author thought the AT101 was the most potent agent to induce the expression of NOXA, but from the results of Fig.1, S1 seems more efficient, especially in Miapaca2, PC3 and NB4. How did S1 induce the level of NOXA?
Answer: We have included references to our prior research demonstrating that S1 induces NOXA through the generation of reactive oxygen species (ROS) that leads to the unfolded-protein response (UPR). While S1 induces more NOXA protein after 6 h compared to AT101 in NB4, PC3 and MiaPaca2 cells, we have included data (Figure 5E) that compares the stabilization of NOXA protein by AT101 and S1. In NB4 cells, AT101 resulted in much more NOXA protein 24 h after its removal than S1. However, in discussion of Figure 1, please note that in addition to the generally higher induction by AT101, we had also written that: “we decided to focus on AT101 in subsequent studies as it has the most extensive clinical testing to date compared to the other compounds.” (Lines 100-101, 245-247, 268-269, and 351)
- In Fig.2D, BAPTA treatment increased the level of NOXA and MCL1 in HCT116. Did Aristol treatment also increase the level of MCL1?
Answer: The review is in error as Fig. 2D shows that BAPTA AM prevented the accumulation of NOXA and MCL1 protein rather than increasing it. We only probed aristolochic acid-treated cells for NOXA, and not MCL1 (Fig 2C), so can not answer the second part of this question. (no change made)
- Both PUMA and NOXA are target genes of p53, however, Bort treatment only increased the level of PUMA but not NOXA. What’s the reason, and whether AT101 treatment could increase the level of p53?
Answer: We have included new data (See Fig. 2G) that shows AT101-induced NOXA is p53-independent. The use of bortezomib related to experiments with PUMA and as discussed in response to reviewer #2, this data has now been deleted. (Lines 129-130 and 149-153)
- MCL1 detection seems required in Fig.3B-D.
Answer: We have revised Fig. 3B-D such that there is consistency. Figure 3 is meant to show how AT101 sensitizes cells to other BH3 mimetics and that this sensitization is NOXA-dependent. MCL1 detection is not required to determine apoptosis as PARP cleavage is sufficient. NOXA expression is required to determine the efficacy of AT101. The changes in MCL1 are not discussed with respect to Fig. 4. (deletion in Figure 3)
- In Fig.1, AT101 could obviously increase the level of NOXA, but this increase seems not obvious in Fig.4A. Why?
Answer: The last lane in each NOXA blot in Fig. 4A shows that AT101 induces a significant amount of NOXA, in contradiction to the reviewer’s question. (no change)
- 5B showed that the level of NOXA mRNA reached a secondary peak at 48h, but not the protein level. NOXA protein level needs to be detected for a longer period of time. And Fig.5A showed the result of AsPC1 which not mentioned in the early part of the paper, did AsPC1 mean PC3?
Answer: The reviewer is correct that the NOXA mRNA appeared to increase again at 48 h, but the level of NOXA protein is still 20-fold above untreated cells, so it is not possible to know how much of the increase in NOXA protein is due to increased stability or the secondary increase in mRNA. (no change)
Answer regarding AsPC-1. We have revised Fig. 5A by deleting AsPC-1 in order to be more consistent throughout the paper. (deletion in figure 5)
- Minor points.In Fig.4, (B through D) should be corrected to B-D (line 184).Line 289, “instead primes” should be corrected to “ instead of priming”.
Answer: The authors have corrected line 184 (now line 216) such that Fig. 4 now says “B-D”. However, line 289 (now line 339) does not need to be corrected as the context of this line requires the use of “instead primes” whereas “instead of priming” would imply the opposite.
- In Fig.5C, cycloheximide should be indicated.
Answer: Fig. 5C now says "time (h) in cycloheximide."
- In first half part and latter part of the paper, the spelling of vinculin is inconformity.
Answer: We have revised all the figures such that any instance of “vinculin” is now replaced by “Vinc”. (corrected throughoout)
Round 2
Reviewer 2 Report
The revised version my Mallick and Eastman addresses my previous criticisms. While I still contend that this is an extension of Dr. Eastman's previous work, it is a well performed study that draws appropriate conclusions from the data.
Reviewer 3 Report
All my concerns have been addressed.